# Collective synthesis of acetylenic pharmaceuticals via enantioselective Nickel/Lewis acid-catalyzed propargylic alkylation

Xihao Chang[1], Jiayin Zhang[1], Lingzi Peng[1] & Chang Guo [1✉]

Chiral acetylenic derivatives are found in many bioactive compounds and are versatile functional groups in organic chemistry. Here, we describe an enantioselective nickel/Lewis acid-catalyzed asymmetric propargylic substitution reaction from simple achiral materials under mild condition. The introduction of a Lewis acid cocatalyst is crucial to the efficiency of the transformation. Notably, we investigate this asymmetric propargylic substitution reaction for the development of a range of structurally diverse natural products. The power of this strategy is highlighted by the collective synthesis of seven biologically active compounds: (−)-Thiohexital, (+)-Thiopental, (+)-Pentobarbital, (−)-AMG 837, (+)-Phenoxanol, (+)-Citralis, and (−)-Citralis Nitrile.

[1] Hefei National Laboratory for Physical Sciences at the Microscale, University of Science and Technology of China, Hefei 230026, China.
✉email: guochang@ustc.edu.cn

The enantiomers of racemic drugs generally alter the pharmacokinetics and pharmacodynamics, which makes the asymmetric synthesis commercially and scientifically interesting[1,2]. Chiral alkynes represent a ubiquitous functional group in many biologically active molecules and exhibit innumerable applications in chemical synthesis, pharmaceuticals, agrochemicals, and material science (Fig. 1a)[3]. However, several challenges are associated with the preparation of a structurally and functionally diverse class of biologically active molecules, such as (i) the development of an economical and concise process to construct an advance core structure in the context of target-oriented synthesis; (ii) large-scale assembly of key building blocks in the drug discovery, development, and testing process; (iii) identification of reaction conditions that afford high enantioselectivities. In particular, barbituric acid alkynes are attractive, as the alkyne site and barbituric acid moiety may easily undergo further deliberate synthetic elaboration for different purposes, and the derivatives of barbiturates have a special place in pharmaceutical chemistry with sedative and anesthetic properties[4]. However, the lack of an efficient enantioselective synthetic method allowed the use of a racemic mixture of chiral barbiturates as drugs (e.g., Thiohexital, Thiopental, and Pentobarbital)[5–7], even though opposite enantiomers of a chiral barbiturate exhibit a different pharmacological profile[8]. Moreover, an alkyne unit in optically active form can be the key component of the medicinal agent AMG 837 as a potent GPR40 agonist for the treatment of type 2 diabetes[9]. Alkyne derivatives also exhibit different olfactory activities in many chiral fragrances such as (+)-Phenoxanol, (+)-Citralis, and (−)-Citralis Nitrile, showing a significant difference in odor threshold between compounds with opposite configuration[10,11]. Owing to their interesting biological properties, challenging structural complicity, and targeted rapid synthesis, the development of modular strategies from a common molecular scaffold for the preparation of large collections of structurally diverse biological compounds in a stereoselective manner is highly desirable[12–14].

Significantly, the stereoselective assembly of chiral alkynes from readily available precursors is a prominent objective in chemical research[3]. We expected the preparation of chiral propargyl-substituted malonate esters[15], which incorporates the requisite functionality for expedient conversion to each of the target bioactive molecules, to be a central element of our design strategy (Fig. 1b). Indeed, various powerful catalysts based on transition metals have been disclosed for the asymmetric propargylic substitution (APS) reactions of terminal propargylic carbonates via the (allenylidene)metal intermediate[16–18], offering a convenient way to install a synthetically versatile alkyne unit onto target molecules[19–21]. Specially, the Wu group developed the copper-catalyzed enantioselective propargylation with propargylic alcohol derivatives bearing a terminal alkyne moiety[22]. The Fu group developed the nickel-catalyzed asymmetric Negishi cross-coupling reaction of propargylic electrophiles with hard nucleophiles via the $Ni^I/Ni^{III}$ catalytic cycle[23,24]. Recently, Kawatsura et al. made seminal contribution in their design of nickel-catalyzed asymmetric Friedel–Crafts propargylation of 3-substituted indoles with propargylic carbonates[25]. However, efficient methods that achieve enantioselective control of the formation of C($sp^3$)–C($sp^3$) bonds adjacent to internal alkynes with malonate derivatives have not been identified[26–32]. In light of the synthetic value of the reaction products, we embarked on the study to address the synthetic challenge and accomplish the enantioselective collective synthesis.

The use of synergistic catalysis has been recognized as a powerful approach for expanding the scope of both strategies and developing valuable transformations[33–41]. Along these lines, our group has reported a dual catalytic strategy that enhances the nucleophilicity of the aldimine esters by chelating them with copper co-catalysts to enable the APS reaction[42]. In this study,

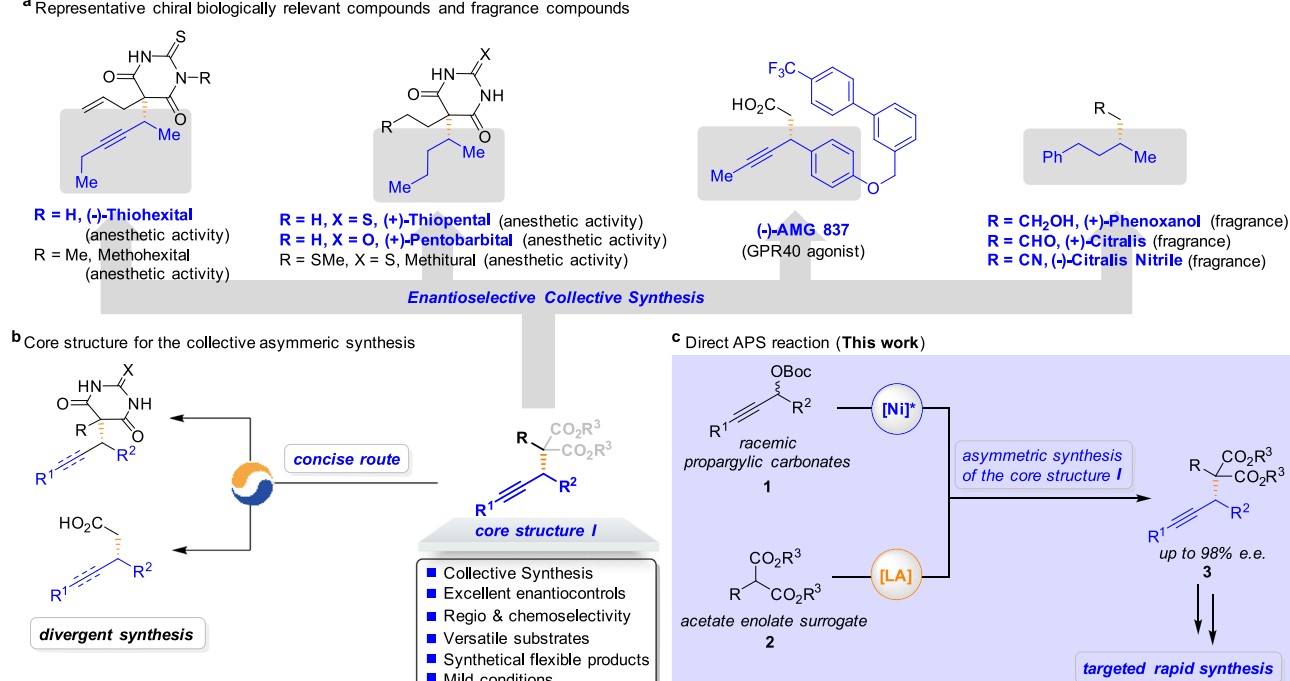

**Fig. 1 Design of the APS reactions for the collective synthesis of seven biological active molecules. a** Enantioselective collective synthesis of (−)-Thiohexital, (+)-Thiopental, (+)-Pentobarbital, (−)-AMG 837, (+)-Phenoxanol, (+)-Citralis, and (−)-Citralis Nitrile. **b** Core structure **I** for the enantioselective collective synthesis. **c** Realization of the Ni/Lewis acid-catalyzed the APS reaction of malonates to enhance the pace of drug discovery and development (this work).

we hypothesized that strategies through the judicious introduction of a Lewis acid co-catalyst to enhance the nucleophilicity of malonates would permit efficient APS reactions (Fig. 1c)[43,44]. In principle, direct nucleophilic substitution may occur stereoselectively at the propargylic carbon affords the corresponding alkylation adduct **3** together with regeneration of the nickel catalyst and the Lewis acid catalyst[45]. Consequently, expanding the scope of the APS reaction concerning both the propargylic electrophiles and the malonate nucleophiles would represent an important advance that will enable and inspire drug synthesis. Here we demonstrate that the synergistic combination of nickel catalysis and Lewis acid catalysis could provide important avenues in the enantioselective collective synthesis of seven biological active molecules: (−)-Thiohexital, (+)-Thiopental,

(+)-Pentobarbital, (−)-AMG 837, (+)-Phenoxanol, (+)-Citralis, and (−)-Citralis Nitrile.

## Results

**Optimization study**. We initiated our investigations via nickel catalysts for the intermolecular alkylation of racemic internal propargylic carbonate **1a** with dimethyl methylmalonate **2a** for the analog synthesis of (−)-Thiohexital (Table 1). Gratifyingly, the (R)-SEGPHOS-Ni(COD)$_2$ complex was found to provide the corresponding product **3a** with 93% e.e. albeit with low conversion (entry 1). To our delight, a variety of Lewis acids were capable of facilitating the title reaction, with ytterbium(III) triflate furnishing the desired product **3a** in 76% yield with 95% e.e. (entries 5 vs 2–4). Encouraged by this result, we next evaluated

**Table 1 Survey on the model reaction conditions.**

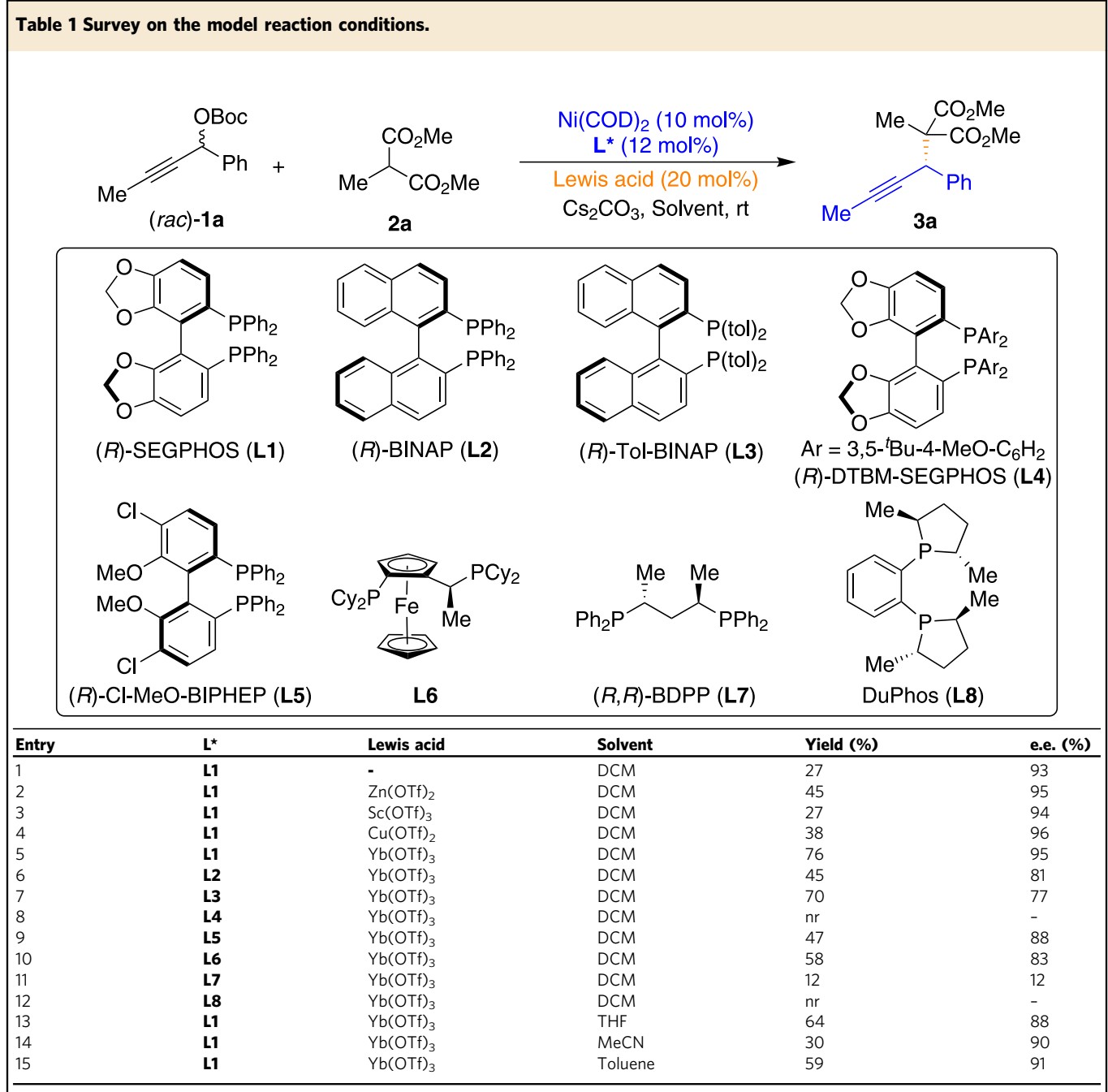

| Entry | L* | Lewis acid | Solvent | Yield (%) | e.e. (%) |
|---|---|---|---|---|---|
| 1 | **L1** | - | DCM | 27 | 93 |
| 2 | **L1** | Zn(OTf)$_2$ | DCM | 45 | 95 |
| 3 | **L1** | Sc(OTf)$_3$ | DCM | 27 | 94 |
| 4 | **L1** | Cu(OTf)$_2$ | DCM | 38 | 96 |
| 5 | **L1** | Yb(OTf)$_3$ | DCM | 76 | 95 |
| 6 | **L2** | Yb(OTf)$_3$ | DCM | 45 | 81 |
| 7 | **L3** | Yb(OTf)$_3$ | DCM | 70 | 77 |
| 8 | **L4** | Yb(OTf)$_3$ | DCM | nr | – |
| 9 | **L5** | Yb(OTf)$_3$ | DCM | 47 | 88 |
| 10 | **L6** | Yb(OTf)$_3$ | DCM | 58 | 83 |
| 11 | **L7** | Yb(OTf)$_3$ | DCM | 12 | 12 |
| 12 | **L8** | Yb(OTf)$_3$ | DCM | nr | – |
| 13 | **L1** | Yb(OTf)$_3$ | THF | 64 | 88 |
| 14 | **L1** | Yb(OTf)$_3$ | MeCN | 30 | 90 |
| 15 | **L1** | Yb(OTf)$_3$ | Toluene | 59 | 91 |

*nr* no reaction, *rt* room temperature. Reactions in this table were conducted with **1a** (0.225 mmol), **2a** (0.15 mmol), Cs$_2$CO$_3$ (0.3 mmol), Ni(cod)$_2$ (10 mol%), **L*** (12 mol%), and Lewis acid (20 mol%) in solvent (2.0 mL) at room temperature for 72 h. e.e. values were determined by high-performance liquid chromatography analysis.

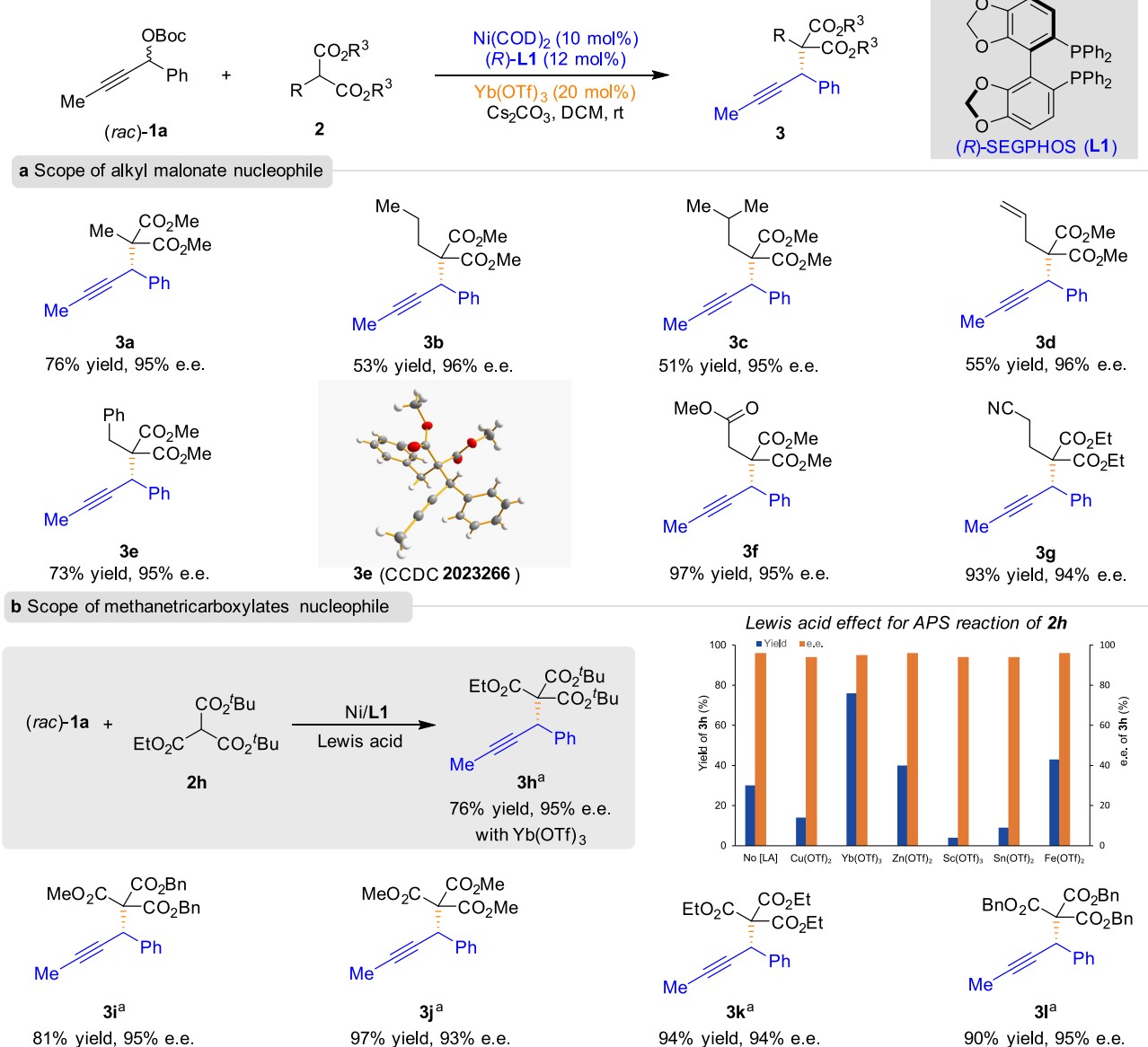

**Fig. 2 Substrate scope of malonate esters. a** Scope of alkyl malonate nucleophile. **b** Scope of methanetricarboxylates nucleophile. Unless otherwise noted, reactions were conducted with **1a** (0.225 mmol), **2** (0.15 mmol), Ni(COD)$_2$ (10 mol%), (*R*)-**L1** (12 mol%), and Yb(OTf)$_3$ (20 mol%) in DCM (2.0 mL) at room temperature for 72 h. $^a$In toluene (2.0 mL).

the APS reaction using different bidentate phosphine ligands in the presence of Ni(COD)$_2$ (entries 6–12). Axially chiral bisphosphine ligands, such as (*R*)-BINAP (**L2**) and (*R*)-Tol-BINAP (**L3**), gave moderate enantiocontrol (entries 6 and 7). However, (*R*)-DTBM-SEGPHOS (**L4**) did not work for this reaction (entry 8). (*R*)-Cl-MeO-BIPHEP ligand (**L5**) afforded **3a** in 88% e.e. (entry 9), whereas JosiPhos ligand (**L6**) gave **3a** in 83% e.e. (entry 10). Furthermore, the use of (*R*,*R*)-BDPP ligand **L7** and DuPhos ligand (**L8**) both gave unsatisfied results (entries 11 and 12). In addition, screening of different solvents gave no improvement in the yield or enantioselectivity of the reaction (entries 13–15).

With the optimized reaction conditions in hand, we then explored the generality of the malonate nucleophiles (Fig. 2). Various substituted malonate esters **2** reacted smoothly with racemic propargylic carbonate **1a** to furnish the propargylic substitution adducts **3** in moderate to good yields with excellent e.e., thereby showing the potential utility of the present reaction

in various organic syntheses (**3a–3g**). The absolute configurations of **3e** were assigned by X-ray single-crystal diffraction study, and the configurations of all other examples were assigned analogously. The reactions of malonates bearing alkene and cyano moieties also furnished the corresponding products in good yields with excellent enantioselectivities (**3d** and **3g**). Gratifyingly, trialkyl methanetricarboxylate **2h** could also be applied via the APS reactions, and the desired product could be obtained in 76% yield and 95% e.e. (**3h**). No improvement in yield was observed for the reaction by screening different Lewis acids. Moreover, a variation of the ester group of methanetricarboxylates was well-tolerated under the current reaction conditions, leading to the corresponding products in good yields with excellent enantioselectivity in all cases (**3i–3l**).

The generality of the reaction concerning the racemic propargylic carbonates **1** was also investigated under the optimized reaction conditions (Fig. 3). A diverse array of propargylic carbonates (**1**) with a variety of functional groups

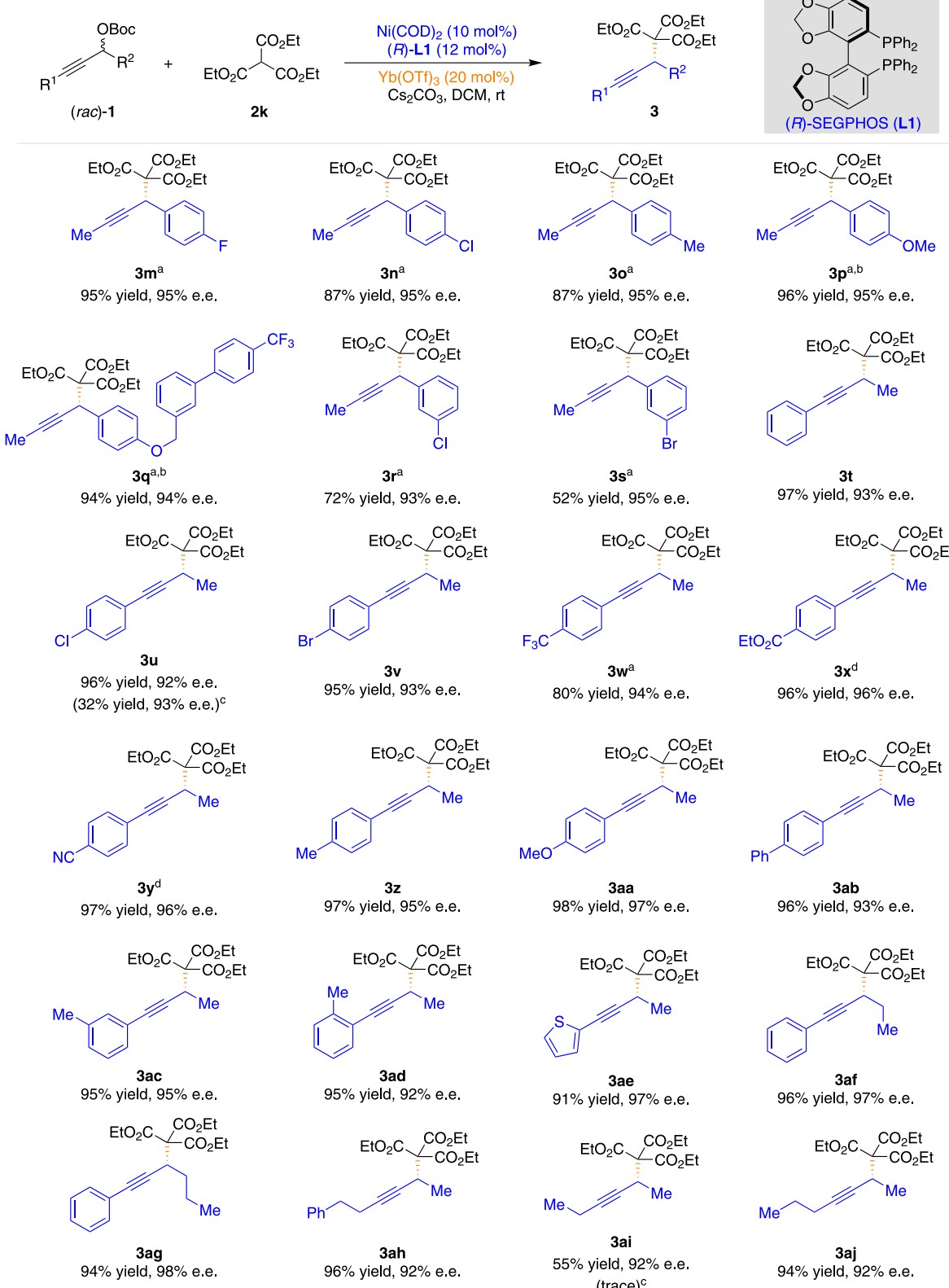

**Fig. 3 Substrate scope of various racemic propargylic carbonates.** [a]In toluene (2.0 mL). [b]In the absence of Lewis acid. [c]The result for the reaction in the presence of Cu(OTf)$_2$ (20 mol%) as the Lewis acid under otherwise identical conditions is given in parentheses. [d]In MeCN (2.0 mL).

(fluoro, chloro, bromo, methyl, and methoxy) on the benzene ring at the propargylic carbon performed well in the APS reaction, and the corresponding products were isolated in high yields and high enantioselectivities (**3m**–**3s**). Propargylic carbonates (**1**) bearing neutral (**3t**, **3ab**), electron-poor (**3u**–**3y**), electron-rich (**3z**–**3aa**), *meta*-(**3ac**), and *ortho*-(**3ad**) substituted arenes were well-tolerated and resulted in excellent levels of enantioselectivities ranging from 92 to 97% e.e. (**3t**–**3ad**).

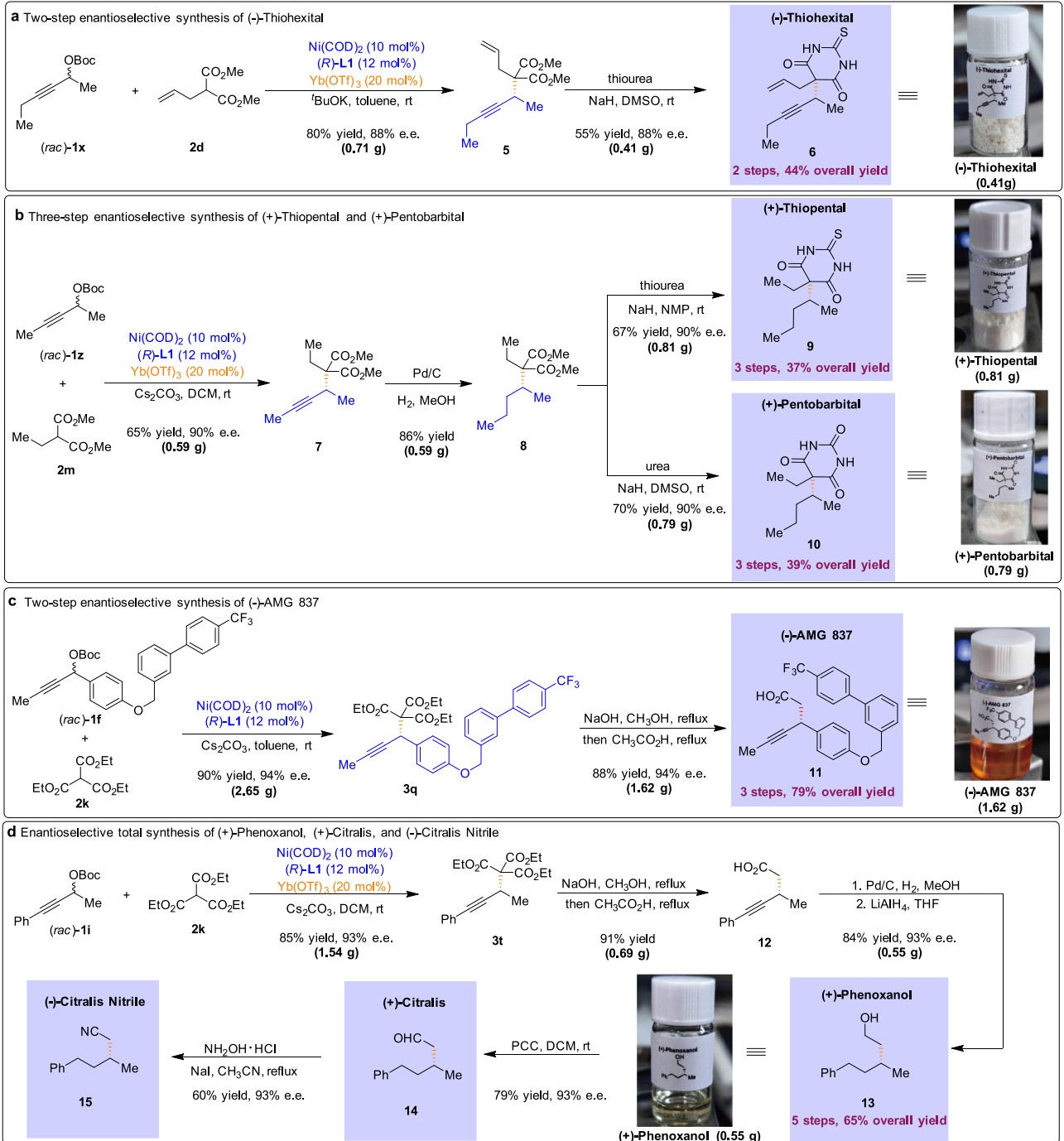

**Fig. 4 Application to chemical synthesis and collective synthesis of seven biologically active compounds. a** Our APS reaction was applied to the two-step enantioselective synthesis of (−)-Thiohexital. **b** The current methodology could allow for the rapid and enantioselective production of (+)-Thiopental and (+)-Pentobarbital. **c** The feasibility of the APS reaction was evaluated in the total synthesis of (−)-AMG 837. **d** The APS reaction provided facile access to (+)-Phenoxanol, (+)-Citralis, and (−)-Citralis Nitrile.

Notably, this method was compatible with the propargylic carbonate bearing the thiophene substituent, giving the desired product in 91% yield with 97% e.e. (**3ae**). Propargylic carbonates bearing various alkyl substituents on the propargylic carbon could also be tolerated without losses in reaction efficiency or enantiocontrol, thus providing opportunities for further elaboration of the products (**3af** and **3ag**). Remarkably, this method was also compatible with dialkyl substituted propargylic carbonates, giving the desired products in good yields and excellent enantioselectivities (**3ah**–**3aj**).

The utility of the present method is highlighted through the enantioselective collective concise synthesis of biologically active compounds (Fig. 4). Specifically, barbiturates are featured prominently in clinically useful pharmaceutical and the configuration of the barbiturates is often crucial for their anesthetic activity[8]. Therefore, concise methods for the synthesis of barbiturate derivatives in high stereochemical purity are particularly valuable. Here we report the development of the APS reaction and its application to the catalytic enantioselective total synthesis of (−)-Thiohexital. The feasibility of this reaction was

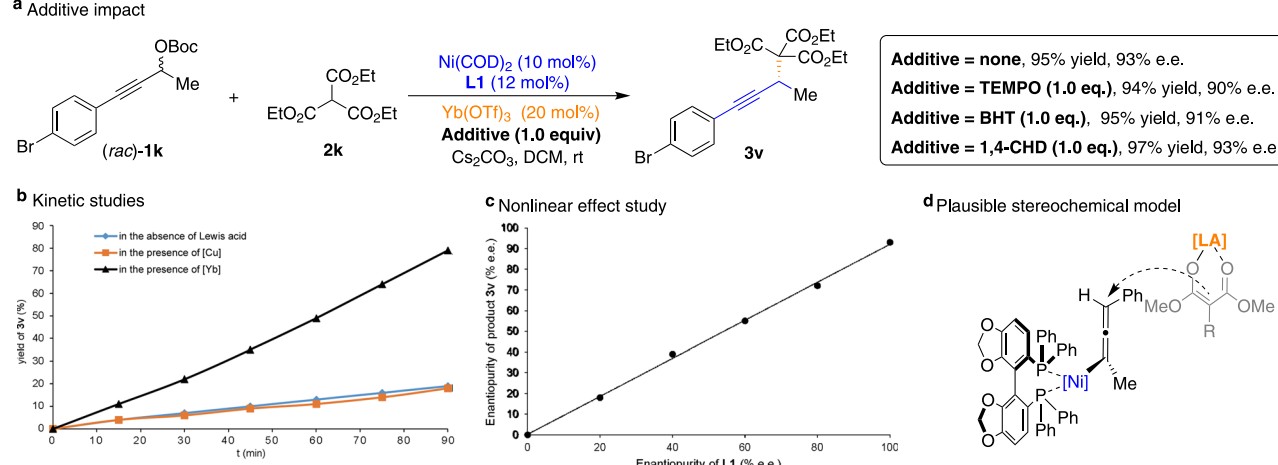

**Fig. 5 Mechanistic studies. a** Radical scavenger experiments. **b** Kinetic studies. The dependence of the reaction rate by varying the Lewis acid additives. **c** Nonlinear relationship between the optical activity of phosphine ligand **L1** and propargylated adduct **3v**. **d** Plausible stereochemical model. TEMPO 2,2,6,6-tetramethylpiperidinyloxy, BHT butylated hydroxytoluene, 1,4-CHD 1,4-cyclohexadiene.

first evaluated in the context of a route to enantioenriched Thiohexital according to the following sequence (Fig. 4a). The crucial propargylic step was accomplished with the use of nickel-phosphine (**L1**) catalyst in the presence of Yb(OTf)$_3$ co-catalyst, forming the propargylated adduct **5** in 80% yield. Condensation of **5** with thiourea in dimethyl sulfoxide (DMSO) containing sodium hydride afforded the desired barbituric acid (−)-Thiohexital with 88% e.e. in 55% yield. In this context, we have completed the catalytic enantioselective total synthesis of (−)-Thiohexital in 2 steps in 44% overall yield (410 mg scale).

To further showcase the synthetic utility of the process, we applied the APS reaction to the 3-step total synthesis of (+)-Thiopental and (+)-Pentobarbital (Fig. 4b). We test the practicability of the described process by conducting the reaction on a large-scale furnish the corresponding adduct **7** in good yield. Hydrogenation of **7** using catalytic Pd/C in methanol under 1 atm H$_2$ led to the reduction of the alkyne group, efficiently providing the saturated ester **8** in 86% yield. Subsequently, the treatment of **8** with thiourea or urea furnished the desired (+)-Thiopental (0.81 g scale in 37% overall yield and 90% e.e.) and (+)-Pentobarbital (0.79 g scale in 39% overall yield and 90% e.e.), respectively.

Based on the strategy of collective total synthesis, we next turned to the facile construction of a potent GPR40 receptor agonist (−)-AMG 837 (Fig. 4c)[9]. Propargylic carbonate **1f** was subjected to the APS reaction on a large scale, and the corresponding propargylated adduct **3q** was obtained in 90% yield with excellent levels of enantioinduction (94% e.e.). Subsequent chemoselective hydrolysis of esters, followed by decarboxylation of the corresponding acids, furnished (−)-AMG 837 in 79% overall yield (1.62 g scale).

Application of our Ni/Lewis acid catalytic strategy allowed access to fragrance compounds, (+)-Phenoxanol, (+)-Citralis, and (−)-Citralis Nitrile, while their enantiomers have been known to exhibit different characteristics (Fig. 4d)[11]. The key APS reaction was accomplished with the use of co-catalysts, forming the adduct **3t** in 85% yield and with excellent e.e. (1.54 g scale). Hydrolysis of esters and decarboxylation of the corresponding acids gave rise to the desired acid **12** in 91% yield (0.69 g scale). Gratifyingly, hydrogenation of **12** in the presence of a catalytic amount of Pd/C, followed by LiAlH$_4$ reduction, generated the corresponding perfume ingredient (+)-Phenoxanol

with 93% e.e. in 65% overall yield (0.55 g scale). Oxidation of alcohol **13** was used in the synthesis of (+)-Citralis in high yield without loss of enantiopurity. Finally, the treatment of (+)-Citralis with hydroxylammonium chloride in the presence of sodium iodide furnished (−)-Citralis Nitrile in 60% yield.

We thus conducted some additional mechanistic investigation of the Ni/Yb-catalyzed APS reactions (Fig. 5). As shown in Fig. 5a, we observed the addition of radical scavengers (TEMPO and BHT) or H-atom donor (1,4-cyclohexadiene) displayed little effects on the outcome of the reaction, thereby ruling out the possibility of a radical mechanism. To gain a better understanding of the mechanistic details of this process, we monitored an analogous kinetic study by varying the Lewis acid additives via $^1$H NMR spectroscopy, indicating that Yb(OTf)$_3$ is superior to other Lewis acid sources in terms of increasing the reaction rate and ultimately the yield (Fig. 5b). Furthermore, we performed a nonlinear effect study to determine the relationship between the e.e. of the SEGPHOS (**L1**) and that of the generated product **3v**[46]. Six reactions were performed using a catalyst with different levels of enantiopurity (racemic, 20%, 40%, 60%, 80%, and >99% e.e). Indeed, the nonlinear effect study revealed a linear relationship between the e.e. of the **3v** and the enantiopurity of phosphine ligand **L1** (Fig. 5c), indicating one molecule of phosphine ligand is most likely involved in the active nickel species. All these observations suggest that the cooperative mechanism is responsible for the high catalytic activity of this system in the Ni/Yb-catalyzed APS reactions. In addition, a plausible transition-state model has been proposed to explain the stereochemical outcome of the reaction (Fig. 5d). Because of the steric repulsion between Lewis acid-bound enolate species and phenyl rings of the chiral phosphine ligand, the direct nucleophilic substitution may occur selectively at the propargylic carbon and afford the corresponding alkylation adduct **3** with observed stereochemistry.

In summary, we have established that the use of Lewis acid could improve the activity of nickel-catalyzed APS reactions of malonates with racemic propargylic carbonates. This urgent transformation proceeds under mild conditions display the remarkable scope and show reliable templates for the enantioselective collective total synthesis. As anticipated, the value of this operationally simple protocol has been highlighted in the process scale production of seven biologically active compounds: (−)-Thiohexital, (+)-Thiopental, (+)-Pentobarbital, (−)-AMG

837, (+)-Phenoxanol, (+)-Citralis, and (−)-Citralis Nitrile. Remarkably, this practical protocol provides opportunities to enhance the pace of drug discovery and development.

## Methods

**Synthesis of 3**. In a nitrogen-filled glove box, an oven-dried 10 mL Schlenk tube equipped with a stir bar was charged with Ni(COD)$_2$ (4.1 mg, 0.015 mmol), **L1** (11.0 mg, 0.018 mmol), and stirred in DCM (2 mL) for about 15 min at room temperature. Then propargylic carbonate (0.225 mmol), malonate (0.15 mmol), Yb (OTf)$_3$ (18.6 mg, 0.03 mmol), and Cs$_2$CO$_3$ (97.7 mg, 0.3 mmol) were added to the tube subsequently under nitrogen atmosphere. The final solution was stirred for about 72 h at room temperature until complete consumption of the malonate (monitored by TLC). The solution was diluted with DCM and washed with water. The aqueous phase was then extracted three times with DCM. The combined organic phase was dried over anhydrous MgSO$_4$ and concentrated under reduced pressure. The residue was purified by silica gel chromatography to afford the desired asymmetric product **3**.

Full experimental procedures are provided in the Supplementary Information.

## Data availability

The authors declare that the data supporting the findings of this study are available within the article and Supplementary information file, or from the corresponding author upon reasonable request. Crystallographic parameters for compounds **3e** are available free of charge from the Cambridge Crystallographic Data Centre (www.ccdc.cam.ac.uk/data_request/cif) under CCDC 2023266.

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

## Acknowledgements

The authors acknowledge financial support from the National Natural Science Foundation of China (grant no. 21702198 and 21971227), the Anhui Provincial Natural Science Foundation (grant no. 1808085MB30), and the Fundamental Research Funds for the Central Universities (WK2340000090).

## Author contributions

X.C. and C.G. conceived and designed the study, and wrote the paper. X.C., J.Z., and L.P. performed the experiments and analyzed the data. C.G. directed the project. All authors discussed the results and commented on the manuscript.

## Competing interests

The authors declare no competing interests.
