## [Peer Review File · Nature Communications]

REVIEWER COMMENTS

Reviewer #1 (Remarks to the Author):

In this manuscript, Chang, Guo and co-workers reported enantioselective propargylic alkylation catalyzed by chiral nickel complexes, wherein racemic internal propargylic carbonates were employed as reaction components. While terminal alkyne propargylic carbonates were typically used for asymmetric catalyzed propargylic substitution in literature. Although nickel catalyzed propargylic components with nucleophiles have been reported previous (J. Am. Chem. Soc. 2008, 130, 38, 12645 and related publications, not cited in this manuscript?), as described in this submitted manuscript it's interesting that the presence of catalytic amount of Lewis acid as a co-catalyst could substantially improve the performance the reaction. Furthermore this methodology was successfully applied in asymmetric collective synthesis of natural products and biologically active compounds. In this context, I suggest the publication of this manuscript on Nat. Commun., and issues listed below should be addressed properly.

- 1) The dual-catalysis strategy presented in this manuscript is interesting, but we found that it's difficult to repeat this reaction in our lab. Could the authors provide more details in SI about the reagents (catalysts or additives) employed in this study?
- 2) In all cases, active methyne nucleophiles were employed for this type of reactions. Can simple methylene nucleophiles, such as dialkyl malonate, Meldrum's acid etc., be employed for this reaction?
- 3) It would be helpful for design of new reactions adopt similar strategy if a transition model could be included in the discussion section to explain the chiral induction of this reaction.

Reviewer #2 (Remarks to the Author):

In the manuscript, the authors described a synergistic Ni/Lewis acid-catalyzed asymmetric propargylic substitution of propargylic carbonates with malonates. Although the similar strategy has been employed in the cooperative Ni/Cu-catalyzed propargylic alkylation by the same group (ACIE 2020, 59, 14270), the present work represents a new insight in the Ni-catalyzed asymmetric propargylic substitution, thus reinforcing the use of this strategy in the construction of complex chiral frameworks. The reaction showed broad substrate scope and gave the substitution products in high yields and enantioselectivities. The utility of this reaction is well demonstrated by the synthesis of some biologically active compounds. Furthermore, the manuscript is well written, and the SI is elaborately prepared. This reviewer therefore recommends this manuscript for the publication in Nat. Commun. After the minor revision as follows,

1. The ee-values of compounds 5 and 7 should be added in Figure 4.
2. The number of malonates seems some confusing, in my suggestion, in Fig. 2a, malonates 2 represent 2a-2g, 2b in the original manuscript should be 2h, and so on.
3. The reaction required too long time (72 h), whether did the authors try to improve the reaction efficiency?
4. How about the substitution with the terminal propargylic esters?

Reviewer #3 (Remarks to the Author):

Guo and co-workers describe the development of the asymmetric propargylic alkylation of propargylic alcohol derivatives bearing an internal alkyne group with malonates by nickel/Lewis acid cooperative catalysis and its application to the collective synthesis of a series of biologically active compounds. The authors revealed that Ni(cod)₂/Yb(OTf)₃ cooperative catalyst system promoted the propargylic substitution of internal alkyne-substituted propargylic carbonates with malonate analogues under mild reaction conditions to give the corresponding alkylation product in good yields with excellent enantioselectivity. This is the first example for the asymmetric propargylic substitution of the substrate class using malonates as a carbon nucleophile. The asymmetric cooperative catalysis enabled novel and efficient access to biologically active

compounds such as (-)-Thiohexital, (+)-Thiopental, (+)-Pentobarbital, (-)-AMG 837, (+)-Phenoxanol, (+)-Citralis, and (-)-Citralis nitrile. The authors also examined the mechanistic study including radical-trapping experiments, kinetic studies, and nonlinear effect study, indicating that an active nickel species derived from one molecular phosphine ligand would involve in the transformation with the aid of the Lewis acid as a co-catalyst.

In this study, the authors showed a novel type of reaction in the field of the catalytic asymmetric propargylic substitution. Furthermore, the synthetic utility of the asymmetric catalysis was clearly demonstrated by the collective synthesis of useful compounds. However, although this study seems to be appropriate for a publication in Nature Communications, the reconsideration and additional experiments are required in terms of the substrate scope, the reaction mechanism, and so on. I show the questions and comments for the manuscript and the supporting information as follows.

Main manuscript:

1. Line 43–46, page 1 to 2, "In recent years ~ onto target molecules." Description about the detailed background in the field of the catalytic asymmetric propargylic substitution of both terminal and internal alkyne-substituted propargylic alcohol derivatives is necessary for explaining the novelty of the present study. In particular, ref.18 and ref.20 should be appropriately follow in the main manuscript, because these research precedents are relevant to the present work.
2. Line 75–77, page 3, "Encouraged by this result, ~ the best choice." Point by point discussion is required for the effect of chiral bisphosphine ligands in this catalysis. I think the choice of ligands clearly affected the catalytic performance.
3. In Table 1, the authors initially used Cs₂CO₃ as a base. I cannot follow the reason why Cs₂CO₃ was used as the base. It is better to mention about the choice of the base employed.
4. In Figure 2, there is no example for the use of simple dialkyl malonates such as diethyl malonate and dimethyl malonate. The authors should examine the reaction using these carbon nucleophiles and show the results whether it is good or not.
5. Line 99–101, page 4, "A diverse ~ the reaction (3m-3aa)." The results observed in the scope of propargylic carbonates should discuss carefully. I cannot follow the applicability and limitation of the present asymmetric catalysis.
6. In Figure 3, I feel that the scope of propargylic carbonates seems to be insufficient for supporting the functional group tolerance of the protocol. There is no example for the application of the substrates bearing electron-withdrawing substituents, such as trifluoromethyl, ester, cyano groups, on their aromatic rings. In addition, propargylic carbonates having heteroaromatic rings should be tested, because the structural motif is ubiquitous in biologically active compounds.
7. In the scope of malonates and propargylic carbonates as shown in Figures 2 and 3, long reaction time (72 h) is required for the completion of the reaction. This is my opinion, the comments about the long reaction time in this reaction would be helpful for readers.
8. In Figure 4, Scheme a and b. There is no information about the enantiomeric excess for compounds 5, 7, and 8. These ee values should be determined, because the condensation step might be a cause of a decrease of the enantiopurity of the final products 6, 9, and 10.
9. In the mechanistic study, the exact role of Yb(OTf)₃ is unclear. In the introduction, the authors mentioned that the Lewis acid would enhance the reactivity of the malonate nucleophile. I think the Lewis acid would also act as an activator of the leaving group on the propargylic carbonates.

Supporting information:

1. Page 1, General information. Please include information about the measurement of the optical rotation value.
2. Page 1, General information. There is no information about the synthetic procedure of the

propargylic carbonates. Synthetic procedure and characterization data of new compounds should be added in the supporting information.

3. Characterization data of IR and melting point for new compounds should be added in the supporting information. In particular, IR measurements would provide useful information about the existing of alkyne, ester, cyano groups in the products.

4. Page 20, X-ray crystallography data. The value of Flack parameter should be included in the table. Furthermore, clear explanation for the determination of the absolute stereochemistry of 3e seems to be required based on Flack parameter, because product 3e does not include heavy atoms such as bromine and chlorine.

Responses to Reviewer Comments

for

Collective synthesis of acetylenic pharmaceuticals via enantioselective Nickel/Lewis acid-catalyzed propargylic alkylation

Xihao Chang, Jiayin Zhang, Lingzi Peng, Chang Guo*

Manuscript number: NCOMMS-20-36262-T

Reply to comments by Reviewer 1

We appreciate Reviewer 1 for the favorable comments and many helpful suggestions!

- (1) In this manuscript, Chang, Guo and co-workers reported enantioselective propargylic alkylation catalyzed by chiral nickel complexes, wherein racemic internal propargylic carbonates were employed as reaction components. While terminal alkyne propargylic carbonates were typically used for asymmetric catalyzed propargylic substitution in literature. Although nickel catalyzed propargylic components with nucleophiles have been reported previous (J. Am. Chem. Soc. 2008, 130, 38, 12645 and related publications, not cited in this manuscript?), as described in this submitted manuscript it's interesting that the presence of catalytic amount of Lewis acid as a co-catalyst could substantially improve the performance the reaction. Furthermore this methodology was successfully applied in asymmetric collective synthesis of natural products and biologically active compounds. In this context, I suggest the publication of this manuscript on Nat. Commun., and issues listed below should be addressed properly.

Answer: We appreciate Reviewer 1 for the favorable comments and helpful suggestions! Those comments are greatly valuable and helpful for revising and improving our paper. We have made all the necessary amendments as suggested in our revised manuscript and revised supporting information. Furthermore, we have cited the above mentioned literature in ref. 23 in our revised manuscript: (Smith, S. W. & Fu, G. C. Nickel-catalyzed asymmetric cross-couplings of racemic propargylic halides with arylzinc reagents. *J. Am. Chem. Soc.* **130**, 12645–12647 (2008).)

(2) The dual-catalysis strategy presented in this manuscript is interesting, but we found that it's difficult to repeat this reaction in our lab. Could the authors provide more details in SI about the reagents (catalysts or additives) employed in this study?

Answer: We appreciate Reviewer 1 for the favorable comments. According to the reviewer's question on reaction repeatability, we provided more details for the alkylation reactions and attached the corresponding TLC plates (propargylated product **3a** and **3v**, **TLC visualization reagents: aqueous alkaline KMnO₄ solution**). **It is worth noting that the polarity of the substrates **2** and the products **3** in the reactions are very close**, which will cause difficulties in monitoring the reaction process. Indeed, we found that dichloromethane is the best choice of eluent for the separation of products **3**. We have added more details according to the detection and the photographic guide for TLC plates in our revised supporting information (Fig. S3).

(3) In all cases, active methyne nucleophiles were employed for this type of reactions. Can simple methylene nucleophiles, such as dialkyl malonate, Meldrum's acid etc., be employed for this reaction?

Answer: We have tried simple methylene nucleophiles (dialkyl malonate and Meldrum's acid) for this reaction. However, none of them could undergo the propargylated reaction, indicating that the substituent of malonate esters is crucial for the reaction to succeed. We have included these results in our revised supporting information (Page S5).

(4) It would be helpful for design of new reactions adopt similar strategy if a transition model could be included in the discussion section to explain the chiral induction of this reaction.

Answer: We propose the possible reaction pathways and stereochemical models as shown in our revised supporting information (Fig. S2). Our transformation initiates with the generation of the electrophilic allenylnickel species **I** via decarboxylation of the propargylic carbonates **1a** with nickel complexes. Meanwhile, the challenging asymmetric propargylation might be enhanced through the introduction of a Lewis acid cocatalyst via in situ deprotonation of acidic hydrocarbons providing access to the generation of nucleophilic enolate species. During the subsequent reaction of allenylnickel species **I** and enolate species, direct nucleophilic substitution may occurs selectively at the propargylic carbon and affords the corresponding alkylation adduct (*R*)-**3a** as the major stereoisomer through the favored **TS I**.

Reply to comments by Reviewer 2

We appreciate Reviewer 2 for the favorable comments and many helpful suggestions!

- (1) In the manuscript, the authors described a synergistic Ni/Lewis acid-catalyzed asymmetric propargylic substitution of propargylic carbonates with malonates. Although the similar strategy has been employed in the cooperative Ni/Cu-catalyzed propargylic alkylation by the same group (ACIE 2020, 59, 14270), the present work represents a new insight in the Ni-catalyzed asymmetric propargylic substitution, thus reinforcing the use of this strategy in the construction of complex chiral frameworks. The reaction showed broad substrate scope and gave the substitution products in high yields and enantioselectivities. The utility of this reaction

is well demonstrated by the synthesis of some biologically active compounds. Furthermore, the manuscript is well written, and the SI is elaborately prepared. This reviewer therefore recommends this manuscript for the publication in Nat. Commun. After the minor revision as follows,

Answer: We appreciate Reviewer 2 for the favorable comments concerning our manuscript! Those comments are greatly valuable and helpful for revising and improving our paper. We have made all the necessary amendments as suggested in our revised manuscript and revised supporting information.

(2) The ee-values of compounds 5 and 7 should be added in Figure 4.

Answer: As the suggestion of Reviewer 2, we have provided the ee-values of compounds 5 (80% yield, 88% e.e.) and 7 (65% yield, 90% e.e.) in the revised manuscript (Fig. 4) and supporting information.

(3) The number of malonates seems some confusing, in my suggestion, in Fig. 2a, malonates 2 represent 2a-2g, 2b in the original manuscript should be 2h, and so on.

Answer: As the suggestion of Reviewer 2, we have renumbered the malonate derivatives 2 in the revised manuscript. At the same time, the renumbered malonate derivatives 2 are also displayed in our revised supporting information.

(4) The reaction required too long time (72 h), whether did the authors try to improve the reaction efficiency?

Answer: Initially, the the (*R*)-SEGPHOS-Ni(COD)₂ complex was found to provide the corresponding product 3a albeit with low conversion in the absence of Lewis acid additive (Table 1). To our delight, a variety of Lewis acids were capable of facilitating the title reaction, with ytterbium(III) triflate furnishing the desired product 3a in 76% yield with 95% e.e. (Table 1, entries 5 vs 2-4). We have found that the Lewis acid co-catalysts could improve the reaction efficiency. Yb(OTf)₃ can effectively increase the reaction rate through the kinetic study in the manuscript (Fig. 5b). In addition, we evaluated a series of bases, and Cs₂CO₃ was proven to be the best choice. Moreover, according to the TLC plates of the propargylic alkylations in Fig. S3 (**TLC visualization reagents: aqueous alkaline KMnO₄ solution**), it can be found that the polarities of the product 3 and the malonate derivatives 2 are very close. If the reactions are not complete, it will be difficult to separate the corresponding products 3. Therefore, the above mentioned reaction time are necessary.

(5) How about the substitution with the terminal propargylic esters?

Answer: As the suggestion of Reviewer 2, terminal propargylic ester was investigated under the optimized reaction conditions. Unfortunately, our catalyst system did not work for the reaction of propargylic carbonate with terminal alkyne group.

Reply to comments by Reviewer 3

We appreciate Reviewer 3 for the favorable comments and many helpful suggestions!

- (1) Guo and co-workers describe the development of the asymmetric propargylic alkylation of propargylic alcohol derivatives bearing an internal alkyne group with malonates by nickel/Lewis acid cooperative catalysis and its application to the collective synthesis of a series of biologically active compounds. The authors revealed that Ni(cod)₂/Yb(OTf)₃ cooperative catalyst system promoted the propargylic substitution of internal alkyne-substituted propargylic carbonates with malonate analogues under mild reaction conditions to give the corresponding alkylation product in good yields with excellent enantioselectivity. This is the first example for the asymmetric propargylic substitution of the substrate class using malonates as a carbon nucleophile. The asymmetric cooperative catalysis enabled novel and efficient access to biologically active compounds such as (-)-Thiohexital, (+)-Thiopental, (+)-Pentobarbital, (-)-AMG 837, (+)-Phenoxanol, (+)-Citralis, and (-)-Citralis nitrile.

The authors also examined the mechanistic study including radical-trapping experiments, kinetic studies, and nonlinear effect study, indicating that an active nickel species derived from one molecular phosphine ligand would involve in the transformation with the aid of the Lewis acid as a co-catalyst.

In this study, the authors showed a novel type of reaction in the field of the catalytic asymmetric propargylic substitution. Furthermore, the synthetic utility of the asymmetric catalysis was clearly demonstrated by the collective synthesis of useful compounds. However, although this study seems to be appropriate for a publication in Nature Communications, the reconsideration and additional experiments are required in terms of the substrate scope, the reaction mechanism, and so on. I show the questions and comments for the manuscript and the supporting information as follows.

Answer: We appreciate Reviewer 3 for the favorable comments and helpful suggestions! Those comments are greatly valuable and helpful for revising and improving our paper. We have made all the necessary amendments as suggested in our revised manuscript and revised supporting information.

- (2) Line 43–46, page 1 to 2, “In recent years ~ onto target molecules.” Description about the detailed background in the field of the catalytic asymmetric propargylic substitution of both terminal and internal alkyne-substituted propargylic alcohol derivatives is necessary for explaining the novelty of the present study. In particular, ref.18 and ref.20 should be

appropriately follow in the main manuscript, because these research precedents are relevant to the present work.

Answer: As the suggestion of Reviewer 3, we have revised the above mentioned sentences in our revised manuscript, as: “Indeed, various powerful catalysts based on transition metals have been disclosed for the asymmetric propargylic substitution (APS) reactions of terminal propargylic carbonates via the (allenylidene)metal intermediate¹⁶⁻¹⁸, offering a convenient way to install a synthetically versatile alkyne unit onto target molecules¹⁹⁻²¹. Specially, The Wu group developed the copper-catalyzed enantioselective propargylation with propargylic alcohol derivatives bearing a terminal alkyne moiety²². The Fu group developed the nickel-catalyzed asymmetric Negishi cross-coupling reaction of propargylic electrophiles with hard nucleophiles via the Ni^I/Ni^{III} catalytic cycle^{23,24}. Recently, Kawatsura et al. made seminal contribution in their design of nickel-catalyzed asymmetric Friedel-Crafts propargylation of 3-substituted indoles with propargylic carbonates²⁵.”. In addition, we have highlighted more details on ref.18 (ref.22 in our revised manuscript) and ref.20 (ref.25 in our revised manuscript) in our revised manuscript.

(3) Line 75–77, page 3, “Encouraged by this result, ~ the best choice.” Point by point discussion is required for the effect of chiral bisphosphine ligands in this catalysis. I think the choice of ligands clearly affected the catalytic performance.

Answer: As the suggestion of Reviewer 3, we have revised the above mentioned sentences in our revised manuscript, as “Encouraged by this results, we next evaluated the APS reaction using different bidentate phosphine ligands in the presence of Ni(COD)₂ (entries 6-12). Axially chiral bisphosphine ligands, such as (*R*)-BINAP (**L2**) and (*R*)-Tol-BINAP (**L3**), gave moderate enantiocontrol (entries 6 and 7). However, (*R*)-DTBM-SEGPHOS (**L4**) did not work for this reaction (entry 8). (*R*)-Cl-MeO-BIPHEP ligand (**L5**) afforded **3a** in 88% e.e. (entry 9), whereas JosiPhos ligand (**L6**) gave **3a** in 83% e.e. (entry 10). Furthermore, the use of (*R,R*)-BDPP ligand **L7** and DuPhos ligand (**L8**) both gave unsatisfied results (entries 11 and 12)”.

(4) In Table 1, the authors initially used Cs₂CO₃ as a base. I cannot follow the reason why Cs₂CO₃ was used as the base. It is better to mention about the choice of the base employed.

Answer: As Reviewer 3’s suggestion on the choice of base, we provided the evaluation results of the bases employed for the propargylic alkylation. We evaluated a series of bases, which displayed remarkable effects on the outcome of the reaction and Cs₂CO₃ as the base was proven to be the best choice. We have included these results in our revised supporting information (Table S1).

Table S1 | Influence of base on the asymmetric propargylic alkylation reaction
Entry	Base	Yield (%)	e.e. (%)
1	-	nr	-
2	K_2CO_3	nr	-
3	Na_2CO_3	nr	-
4	KO^tBu	39	68
5	NaOMe	nr	-
6	NaHMDS	54	79
7	LiHMDS	81	82
8	DIPEA	nr	-
9	Pyridine	nr	-
10	Cs_2CO_3	76	95

Reactions were conducted with **1a** (0.225 mmol), **2a** (0.15 mmol), base (0.3 mmol), $\text{Ni}(\text{COD})_2$ (10 mol%), **L1** (12 mol%) and $\text{Yb}(\text{OTf})_3$ (0.03mmol) in DCM (2.0 mL) at room temperature for 72 h. e.e. values were determined by high-performance liquid chromatography analysis. nr = no reaction. rt = room temperature.

(5) In Figure 2, there is no example for the use of simple dialkyl malonates such as diethyl malonate and dimethyl malonate. The authors should examine the reaction using these carbon nucleophiles and show the results whether it is good or not.

Answer: As the suggestion of Reviewer 3, we have tried simple methylene nucleophiles (diethyl malonate, dimethyl malonate and Meldrum's acid) for this reaction. However, none of them could undergo the propargylated reaction, indicating that the substituent of malonate esters is crucial for the reaction to succeed. We have included these results in our revised supporting information (Page S5).

(6) Line 99–101, page 4, “A diverse ~ the reaction (3m-3aa).” The results observed in the scope of propargylic carbonates should discuss carefully. I cannot follow the applicability and limitation of the present asymmetric catalysis.

Answer: As the suggestion of Reviewer 3, we have revised the above mentioned sentences in our revised manuscript, as “A diverse array of propargylic carbonates (**1**) with a variety of functional groups (fluoro, chloro, bromo, methyl and methoxy) on the benzene ring at the propargylic carbon performed well in the APS reaction, and the corresponding products were isolated in high yields and high enantioselectivities (**3m-3s**). Propargylic carbonates (**1**) bearing neutral (**3t,3ab**), electron-poor (**3u-3y**), electron-rich (**3z-3aa**), *meta*-(**3ac**) and *ortho*-(**3ad**) substituted arenes were well tolerated and resulted in excellent levels of enantioselectivities ranging from 92%-97% e.e. (**3t-**

3ad). Notably, this method was compatible with the propargylic carbonate bearing the thiophene substituent, giving the desired product in 91% yield with 97% e.e. (**3ae**).”.

(7) In Figure 3, I feel that the scope of propargylic carbonates seems to be insufficient for supporting the functional group tolerance of the protocol. There is no example for the application of the substrates bearing electron-withdrawing substituents, such as trifluoromethyl, ester, cyano groups, on their aromatic rings. In addition, propargylic carbonates having heteroaromatic rings should be tested, because the structural motif is ubiquitous in biologically active compounds.

Answer: As the suggestion of Reviewer 3, we tested the suitability of substrates with electron-withdrawing substituents on aromatic rings, such as trifluoromethyl, ester, cyano and heteroaromatic rings. Remarkably, this method was compatible with electron-withdrawing substituted propargylic carbonates, giving the desired products (**3w-3y**) in good yields and excellent enantioselectivities. As expected, propargylic carbonate with heteroaromatic ring was also suitable coupling partner, which leads to the formation of the product **3ae** in 91% yield and 97% e.e.. We have added these results in our revised manuscript.

(8) In the scope of malonates and propargylic carbonates as shown in Figures 2 and 3, long reaction time (72 h) is required for the completion of the reaction. This is my opinion, the comments about the long reaction time in this reaction would be helpful for readers.

Answer: Initially, the the (*R*)-SEGPHOS-Ni(COD)₂ complex was found to provide the corresponding product **3a** albeit with low conversion in the absence of Lewis acid additive (Table 1). To our delight, a variety of Lewis acids were capable of facilitating the title reaction, with ytterbium(III) triflate furnishing the desired product **3a** in 76% yield with 95% e.e. (Table 1, entries 5 vs 2-4). We have found that the Lewis acid co-catalysts could improve the reaction efficiency. Yb(OTf)₃ can effectively increase the reaction rate through the kinetic study in the manuscript (Fig. 5b). In addition, we evaluated a series of bases, and Cs₂CO₃ was proven to be the best choice. Moreover, according to the TLC plates of the propargylic alkylations in Fig. S3 (**TLC visualization reagents: aqueous alkaline KMnO₄ solution**), it can be found that the polarities of the product **3** and the malonate derivatives **2** are very close. If the reactions are not complete, it will be difficult to separate the corresponding products **3**. Therefore, the above mentioned reaction time are necessary.

(9) In Figure 4, Scheme a and b. There is no information about the enantiomeric excess for compounds **5**, **7**, and **8**. These ee values should be determined, because the condensation step might be a cause of a decrease of the enantiopurity of the final products **6**, **9**, and **10**.

Answer: As the suggestion of Reviewer 3, we have provided the e.e.-values of compounds **5** (80% yield, 88% e.e.) and **7** (65% yield, 90% e.e.) in the revised manuscript (Fig. 4) and supporting information. However, we could not determine the e.e. value of compound **8** via HPLC analysis and GC analysis. In addition, we have determined the e.e. values of compounds **5** (80% yield, 88%

e.e.) and **7** (65% yield, 90% e.e.), which proved that the condensation reaction does not reduce the enantiomeric purity to afford corresponding products **6** (55% yield, 88% e.e.), **9** (67% yield, 90% e.e.) and **10** (70% yield, 90% e.e.), respectively.

(10) In the mechanistic study, the exact role of $\text{Yb}(\text{OTf})_3$ is unclear. In the introduction, the authors mentioned that the Lewis acid would enhance the reactivity of the malonate nucleophile. I think the Lewis acid would also act as an activator of the leaving group on the propargylic carbonates.

Answer: Recently, our group has reported a dual catalytic strategy that enhances the nucleophilicity of the aldimine esters by chelating them with copper cocatalysts to enable the APS reaction (ref. 42). The coordination of the copper catalyst to aldimine ester gives rise to the nucleophilic metalated azomethine ylide. In fact, the direct catalytic substitution of propargylic alcohols or propargylic acetates on polarization by the Lewis acid has been reported to generate the incipient propargylic carbocation followed by the nucleophilic addition, which might lead to the corresponding racemic products (ref. 18: Roy, R. & Saha, S. Scope and Advances in the Catalytic Propargylic Substitution Reaction. *RSC Adv.* **8**, 31129-31193 (2018)). Furthermore, Kawatsura et al. reported an asymmetric nickel-catalyzed propargylation of 3-substituted indoles with propargylic carbonates (ref. 25). Considering that this reaction does not involve a $\text{Yb}(\text{OTf})_3$ cocatalyst to generate the nucleophilic enolate species and that only the Ni-catalytic cycle is taking place, we monitored an analogous kinetic study of Lewis acid effect via ^1H NMR spectroscopy to investigate the exact role of $\text{Yb}(\text{OTf})_3$. However, the use of 20 mol% of $\text{Yb}(\text{OTf})_3$ as an additive could lead to the desired product in comparable reaction rate. Therefore, we hypothesized that the judicious introduction of a Lewis acid cocatalyst to enhance the nucleophilicity of malonate derivatives might permit the APS reactions under our optimized reaction conditions. However, an activator of the leaving group on the propargylic carbonates via the Lewis acid could not be ruled out.

Supporting information:

(11) Page 1, General information. Please include information about the measurement of the optical rotation value.

Answer: We have provided information on the measured values of optical rotation in the general information in our revised supporting information.

(12) General information. There is no information about the synthetic procedure of the propargylic carbonates. Synthetic procedure and characterization data of new compounds should be added in the supporting information.

Answer: As the suggestion of Reviewer 3, we have added the information about the synthetic procedure and characterization data of propargylic carbonates and new compounds in the revised supporting information.

(13) Characterization data of IR and melting point for new compounds should be added in the supporting information. In particular, IR measurements would provide useful information about the existing of alkyne, ester, cyano groups in the products.

Answer: As the suggestion of Reviewer 3, we have added the characterization data of IR and melting point for new compounds in the revised supporting information.

(14) Page 20, X-ray crystallography data. The value of Flack parameter should be included in the table. Furthermore, clear explanation for the determination of the absolute stereochemistry of

3e seems to be required based on Flack parameter, because product 3e does not include heavy atoms such as bromine and chlorine.

Answer: As the suggestion of Reviewer 3, we have added the value of Flack parameter for the X-ray crystallography data in the revised supporting information (Table S2). The value of the Flack parameter is close to zero, therefore, the absolute structure given by the structure refinement is likely correct (Parsons, S; Flack, H. D; Wagner, T. Use of intensity quotients and differences in absolute structure refinement. *Acta Cryst.* **2013**, *B69*, 249.).

Table S2. X-ray crystallography data of 3e

3e	
Chemical formula	C ₂₂ H ₂₂ O ₄
Formula weight	350.39
Space group	P1
Z	2
α , Å	8.7530(2)
b, Å	10.4823(2)
c, Å	11.9201(3)
α , °	98
β , °	104
γ , °	114
V, Å ³	932.02(4)
Flack parameter	0.06(7)

REVIEWERS' COMMENTS

Reviewer #1 (Remarks to the Author):

In the revised manuscript most of the questions raised by three reviewers were well addressed. Further experiments were carried out to answer these questions. The manuscript is now suitable for publication. One more minor question: Since that the authors have proposed a reasonable transition model to explain the chiral induction of this reaction in the supporting information, why not include a short discussion of chiral induction in the mechanistic studies section?

Reviewer #2 (Remarks to the Author):

The authors have revised the manuscript according to the reviewers's comments and answered their concerns. I therefore recommend this manuscript for the publication in Nat. Commun..

Reviewer #3 (Remarks to the Author):

The authors adequately respond to questions and comments from reviewers. The revised manuscript and the supporting information include revisions and additional information. The present manuscript would be suitable for a publication in Nature Communications after minor revision for the supporting information as follows.

1. Page 2, General information. Please show a path length for the measurement of the optical rotation value.
2. Please include ^1H , ^{13}C , and ^{19}F NMR spectra for the propargylic carbonates 1a–z.

Responses to Reviewer Comments

for

Collective synthesis of acetylenic pharmaceuticals via enantioselective Nickel/Lewis acid-catalyzed propargylic alkylation

Xihao Chang, Jiayin Zhang, Lingzi Peng, Chang Guo*

Manuscript number: NCOMMS-20-36262A

Reply to comments by Reviewer 1

We appreciate Reviewer 1 for the favorable comments and many helpful suggestions!

- (1) In the revised manuscript most of the questions raised by three reviewers were well addressed. Further experiments were carried out to answer these questions. The manuscript is now suitable for publication. One more minor question: Since that the authors have proposed a reasonable transition model to explain the chiral induction of this reaction in the supporting information, why not include a short discussion of chiral induction in the mechanistic studies section?

Answer: We appreciate Reviewer 1 for the favorable comments and helpful suggestions! As the suggestion of Reviewer 1, we have added the transition model and discussion of chiral induction in the mechanistic studies section in our revised manuscript, as “In addition, a plausible transition-state model has been proposed to explain the stereochemical outcome of the reaction (Fig. 5d). Because of the steric repulsion between Lewis acid-bound enolate species and phenyl rings of the chiral phosphine ligand, the direct nucleophilic substitution may occur selectively at the propargylic carbon and afford the corresponding alkylation adduct **3** with observed stereochemistry.”.

Reply to comments by Reviewer 2

We appreciate Reviewer 2 for the favorable comments and many helpful suggestions!

- (1) The authors have revised the manuscript according to the reviewers's comments and answered their concerns. I therefore recommend this manuscript for the publication in Nat. Commun..

Answer: We appreciate Reviewer 2 for the favorable comments and helpful suggestions!

Reply to comments by Reviewer 3

We appreciate Reviewer 3 for the favorable comments and many helpful suggestions!

- (1) The authors adequately respond to questions and comments from reviewers. The revised manuscript and the supporting information include revisions and additional information. The present manuscript would be suitable for a publication in Nature Communications after minor revision for the supporting information as follows.

Answer: We appreciate Reviewer 3 for the favorable comments and helpful suggestions!

- (2) Page 2, General information. Please show a path length for the measurement of the optical rotation value.

Answer: We have provided information on the path length (1 dm path length cell) for the measurement of the optical rotation value in our revised Supplementary Information.

- (3) Please include ¹H, ¹³C, and ¹⁹F NMR spectra for the propargylic carbonates 1a–z.

Answer: We have provided ¹H, ¹³C and ¹⁹F NMR spectra for the propargylic carbonates 1a–z in our revised Supplementary Information.